


# Technical note: Discharge response of a confined aquifer with variable thickness to temporal nonstationary random recharge processes

**Ching-Min Chang[1], Chuen-Fa Ni[1], We-Ci Li[1], Chi-Ping Lin[2], and I-Hsian Lee[2]**

[1]Graduate Institute of Applied Geology, National Central University, Taoyuan, Taiwan

[2]Center for Environmental Studies, National Central University, Taoyuan, Taiwan

**Correspondence:** Chuen-Fa Ni (nichuenfa@geo.ncu.edu.tw)





**Abstract.** This work develop a transfer function to describe the variation of the
integrated specific discharge in response to the temporal variation of the rainfall event
in the frequency domain. It is assumed that the rainfall-discharge process takes place in
a confined aquifer with variable thickness, and it is treated as nonstationary in time to
represent the stochastic nature of the hydrological process. The presented transfer
function can be used to quantify the variability of the integrated discharge field
induced by the variation of rainfall field or to simulate the discharge response of the
system to any varying rainfall input at any time resolution using the convolution model.
It is shown that with the Fourier-Stieltjes representation approach a closed-form
expression for the transfer function in the frequency domain can be obtained, which
provide a basis for the analysis of the influence of controlling parameters occurring in
the rainfall rate and integrated discharge models on the transfer function.
**1  Introduction**
Quantifying the variability of specific discharge response of an aquifer system to
fluctuations in inflow recharge is essential for efficient groundwater resources
management. However, this requires extensive and continuous hydrological
time-series data, and these data are very often not available in practice. One possible

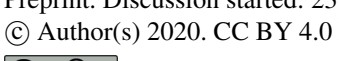



approach (namely, convolution or transfer function approach) to this problem is to
simulate the discharge response by convolution of the time-varying recharge input
with the corresponding impulse response. In convolution models, the aquifer is
regarded as a filter that converts recharge signals into fluctuations of the aquifer head
or discharge. Lumped conceptual-convolution models have been shown to be an
efficient means for the simulation of time series of groundwater levels (e.g., Gelhar,
1974; Molénat et al., 1999; Olsthoorn, 2007; Long and Mahler, 2013; Pedretti et al.,

2016).

Since the impulse response function in the convolution model contains all

information of the system necessary to relate its input to its output, it may be
determined from the analytical solution of the linear system equation governing the
input-output process (e.g., Cooper and Rorabaugh, 1963). Once a suitable impulse
response function can be specified, it allows the simulation of the linear system
response to any varying input at any time resolution.

In this work, a regional-scale flow in a confined aquifer with variable thickness,

which is recharged by rainfall through an outcrop, is analyzed by deriving transfer
functions to characterize the rainfall-discharge process in the frequency domain. The
stochastic analysis of groundwater flow is traditionally based on the assumption of
stationarity of the recharge and discharge processes. However, the hydrologic process
in nature is nonstationary-stochastic (e.g., Christensen and Lettenmaier, 2007; Milly
et al., 2008; Sang et al., 2018). In order to improve the quantification of the natural
recharge-discharge process, the nonstationary rainfall-discharge process is assumed in
this study. The Fourier-Stieltjes representation approach is used to achieve the goal of
this work. The analysis of the results is focused on the influence of controlling
parameters in the rainfall-discharge models on the transfer function.

**2    Problem formulation**

This study regards the entire confined aquifer of variable thickness with stochastic
rainfall recharge and thus stochastic outflow as a single lumped linear system. This
means that the control volume is extended to the scale of an aquifer, so that the flow
variables are integrated in space and only the temporal variability is preserved. In this
way, the output of the system can be represented as a linear combination of the
responses to each of the basic inputs (e.g., Rugh, 1981; Rinaldo & Marani, 1987)
$$Q(t) = \int_0^t \varphi(t,\tau)R(\tau)d\tau, \tag{1}$$

where $Q$ and $R$ denote the output flow rate and the input flow (or recharge) rate of the
system, respectively, and $\varphi$ is the impulse response function of the system. This
implies that once an appropriate impulse response function can be specified on the





aquifer scale, the evaluation of the system response does not require the specification
of a smaller scale heterogeneity.

When using the nonstationary Fourier-Stieltjes representations for the perturbed

quantities of random recharge and outflow discharge processes, namely (e.g.,
Priestley, 1965)

$$r(t) = R(t) - E[R(t)] = \int_{-\infty}^{\infty} \Lambda_r(t;\omega)dZ_\xi(\omega),$$    (2)

$$q(t) = Q(t) - E[Q(t)] = \int_{-\infty}^{\infty} \Lambda_q(t;\omega)dZ_\xi(\omega),$$    (3)

the power spectrum of the mean-removed convolution (1) can be written in the form

$$S_{qq}(t;\omega) = \left|\Lambda_q(t;\omega)\right|^2 S_{\xi\xi}(\omega),$$    (4)

where

$$\Lambda_q(t;\omega) = \int_0^t \varphi(t,\tau)\Lambda_r(\tau;\omega)d\tau.$$    (5)

In Eqs. (2) and (3), $\Lambda_r$ and $\Lambda_q$ are the oscillatory functions (Priestley, 1965) of the
recharge and outflow processes, respectively, $\omega$ is the frequency, $\xi$ is a zero-mean
random stationary forcing process, which generates the variations of the recharge and
thus the output flow processes, with an orthogonal increment $dZ_\xi$. In Eq. (4), $S_{qq}$ and
$S_{\xi\xi}$ represent the power spectra of the processes $q$ and $\xi$, respectively, and $|\Lambda_q|^2$ is
termed the transfer function.

In practice, the interest in many cases resides in evaluating the influence of the



variation of recharge on the variation of the outflow discharge. Equation (4) provides
an efficient way to quantify the variability of the outflow induced by the fluctuations
of the inflow process in the frequency domain, since it relates the fluctuations of an
output time series to those of an input series.
It is worthwhile to mention that for the case of second-order stationary rainfall
processes, the representations of the forms (2) and (3) are reduced, respectively, to
$$r(t) = \int_{-\infty}^{\infty} e^{i\omega t} dZ_r(\omega), \qquad\qquad (6)$$

$$q(t) = \int_{-\infty}^{\infty} \Lambda_q(t;\omega) dZ_r(\omega), \qquad\qquad (7)$$

and correspondingly
$$S_{qq}(t;\omega) = \left| \Lambda_q(t:\omega) \right|^2 S_{rr}(\omega), \qquad\qquad (8)$$

where
$$\Lambda_q(t;\omega) = \int_0^t \varphi(t,\tau) e^{i\omega t} d\tau . \qquad\qquad (9)$$

Equations (1) and (4) reveal that once the transfer function for the linear lumped
system is identified, the first two moments of temporal random discharge fields can be
determined. That is, the transfer function approach provides a basic framework for the
characterization of large-scale flow processes, which may service as a basis for an
efficient management of groundwater resources. Furthermore, Eq. (4) provides
another possible way to identify the aquifer parameters, as it relates the observed





fluctuations of an output discharge process to those of a recharge process in the
frequency domain.

In the following, the focus is on the development of a closed-form expression for

the transfer function for a linear lumped confined flow model, in which the regional
confined aquifer is directly recharged by rainfall in the area corresponding to the high
elevation outcrop.

**3    Theoretical development**

The differential equation describing the transient flow of groundwater in
inhomogeneous isotropic confined aquifers is of the form (e.g., Bear, 1979; de
Marsily, 1986)
$$S_s \frac{\partial}{\partial t} h(\boldsymbol{x},t) = \frac{\partial}{\partial x_i}[K(\boldsymbol{x})\frac{\partial}{\partial x_i} h(\boldsymbol{x},t)] \qquad i = 1, 2, 3,$$    (10)
in which $S_s$ represents the specific storage coefficient of the aquifer, $h = h(\boldsymbol{x},t)$ is the
hydraulic head, $K(\boldsymbol{x})$ is the hydraulic conductivity, and $\boldsymbol{x}$ (= $(x_1,x_2,x_3)$) is the spatial
coordinate vector. Many problems of groundwater flow are regional in nature, with
the horizontal extent of the formation being much larger than the vertical extent. It is
more practical to regard the flow as essentially horizontal. The regional-scale flow
equations can be derived by integrating Eq. (10) along the thickness of the confined



aquifer using the assumption of vertical equipotential surfaces (e.g., Bear, 1979; Bear
and Cheng, 2010).

Integrating Eq. (10) along the $x_3$-axis perpendicular to the confining beds and

using Leibnitz' rule results in
$$S(x_1,x_2)\frac{\partial}{\partial x_i}\tilde{h}(x_1,x_2,t) = \frac{\partial}{\partial x_i}[T(x_1,x_2)\frac{\partial}{\partial x_i}\tilde{h}(x_1,x_2,t)] + T(x_1,x_2)\frac{\partial}{\partial x_i}\ln B(x_1,x_2)\frac{\partial}{\partial x_i}\tilde{h}(x_1,x_2,t), \quad i = 1, 2 \quad (11)$$
where $S(x_1,x_2)$ is the storage coefficient (or storativity) of the aquifer ($= S_sB(x_1,x_2)$),
$B(x_1,x_2) = b_2(x_1,x_2)-b_1(x_1,x_2)$ (an aquifer's thickness), $T(x_1,x_2)$ is the transmissivity of the
aquifer ($=K(x_1,x_2)B(x_1,x_2)$), interpreted as the depth-integrated hydraulic conductivity,
and $\tilde{h}(x_1,x_2,t)$ is the depth-averaged hydraulic head defined as
$$\tilde{h}(x_1,x_2) = \frac{1}{b_2(x_1,x_2)-b_1(x_1,x_2)}\int_{b_1(x_1,x_2)}^{b_2(x_1,x_2)} h(x_1,x_2,x_3,t)dx_3, \quad (12)$$
Equation (11) is derived under the following assumptions: (1) there is no exchange of
leakage fluxes between the confined aquifer and its confining beds in the direction of
$x_3$-axis, and (2) $h(x_1,x_2,b_2,t) \approx \tilde{h}(x_1,x_2,t) \approx h(x_1,x_2,b_1,t)$ (vertical equipotentials; Bear,
1979; Bear and Cheng, 2010). Similarly, when applying Leibnitz' rule to Darcy
equation, the vertically integrated specific discharge in the $x_i$ direction is given by
$$Q_{x_i}(x_1,x_2,t) = -K(x_1,x_2)B(x_1,x_2)\frac{\partial}{\partial x_i}\tilde{h}(x_1,x_2,t) = -T(x_1,x_2)\frac{\partial}{\partial x_i}\tilde{h}(x_1,x_2,t). \quad i = 1, 2 \quad (13)$$

If the regional confined aquifer has nonuniform, unidirectional mean flow in the

direction of $x_1$-axis, but with small flow fluctuations in the direction of $x_1$- and $x_2$-axis
and time-varying recharge at the aquifer outcrop ($x_1 = 0$), the groundwater flow may





be regarded as one-dimensional, so that Eqs. (11) and (13) can be approximated,
respectively, by

$$\frac{S(x)}{\overline{T}}\frac{\partial}{\partial t}\tilde{h}(x,t)=\frac{\partial^2}{\partial x^2}\tilde{h}(x,t)+\frac{\partial}{\partial x}\ln\overline{T}(x)\frac{\partial}{\partial x}\tilde{h}(x,t)+\frac{\partial}{\partial x}\ln B(x)\frac{\partial}{\partial x}\tilde{h}(x,t)+\frac{R(t)}{\overline{T}},\tag{14}$$

$$Q_x(x,t)=-\overline{T}(x)\frac{\partial}{\partial x}\tilde{h}(x,t),\tag{15}$$

where $\overline{T} = \overline{K}B$, $\overline{K}$ represents the spatial average of the hydraulic conductivity,
and R is the recharge rate. Equation (14) can be expressed alternatively as

$$\frac{S_s}{\overline{K}}\frac{\partial}{\partial t}\tilde{h}(x,t)=\frac{\partial^2}{\partial x^2}\tilde{h}(x,t)+2\frac{\partial}{\partial x}\ln B(x)\frac{\partial}{\partial x}\tilde{h}(x,t)+\frac{R(t)}{\overline{K}B(x)}.\tag{16}$$

for the convenient analysis of the effect of the thickness of the aquifer.
In the following analysis, the recharge rate is considered a random function of
time. Equation (15) is then regarded as a stochastic differential equation with a
stochastic input in time and therefore a stochastic output in time. Introduction of
decomposition of the depth-averaged hydraulic head into a mean and a zero-mean
perturbation into Eq. (16) and, after subtracting the mean of the resulting equation
from Eq. (16), the result is the following equation describing the depth-averaged head
perturbation

$$\frac{S_s}{\overline{K}}\frac{\partial}{\partial t}h'(x,t)=\frac{\partial^2}{\partial x^2}h'(x,t)+2\frac{\partial}{\partial x}\ln B(x)\frac{\partial}{\partial x}h'(x,t)+\frac{r(t)}{B(x)\overline{K}},\tag{17}$$

where $h'(x,t)$ is the fluctuations in depth-averaged head.
If it is assumed that the thickness of confined aquifer increase in x-direction in
accordance with (Hantush, 1962; Marino and Luthin, 1982)





$B(x) = \beta e^{\alpha x}$,    (18)
then Eq. (17) becomes
$\dfrac{S_s}{K}\dfrac{\partial}{\partial t}h'(x,t) = \dfrac{\partial^2}{\partial x^2}h'(x,t) + 2\alpha\dfrac{\partial}{\partial x}h'(x,t) + \dfrac{e^{-\alpha x}}{\beta K}r(t)$.    (19)
In Eq. (18), $\beta$ and $\alpha$ are positive geometrical parameters. Furthermore, the outcrop ($x$
$= 0$) and outlet ($x = L$) of the confined aquifer are considered as constant head
boundaries. Since Eq. (19) only quantifies the response of the depth-averaged head to
changes in the recharge rate, the initial and boundary conditions for Eq. (19) may be
represented as follows
$h'(x,0;\omega) = 0$,    (20a)
$h'(0,t;\omega) = 0$,    (20b)
$h'(L,t;\omega) = 0$.    (20c)
The following Fourier-Stieltjes integral representation of a depth-averaged head
process is used to solve Eqs. (19) and (20) for the fluctuations $h'$ in terms of $r$:
$h'(x,t) = \displaystyle\int_{-\infty}^{\infty} \Lambda_h(t;\omega)dZ_\xi(\omega)$,    (21)
where $\Lambda_h$ is the oscillatory function of depth-averaged head process. The resulting
differential equation for the oscillatory functions is found from using Eqs. (2) and (21)
in Eqs. (19) and (20) as
$\dfrac{S_s}{K}\dfrac{\partial}{\partial t}\Lambda_h(x,t;\omega) = \dfrac{\partial^2}{\partial x^2}\Lambda_h(x,t;\omega) + 2\alpha\dfrac{\partial}{\partial x}\Lambda_h(x,t;\omega) + \dfrac{e^{-\alpha x}}{\beta K}\Lambda_r(t;\omega)$.    (22)
with the following conditions:


$\quad \Lambda_h(x,0;\omega)=0,$ (23a)
$\quad \Lambda_h(0,t;\omega)=0,$ (23b)
$\quad \Lambda_h(L,t;\omega)=0.$ (23c)
By solving the above boundary value problem, the oscillatory function of
depth-averaged head process is found to be (see Appendix A)
$\quad \Lambda_h(x,t;\omega)=\dfrac{2}{S_s\beta}\sum\limits_{n=1}^{n=\infty}\dfrac{1-\cos(n\pi)}{n\pi}\exp(-\mu\dfrac{x}{L})\sin(n\pi\dfrac{x}{L})\int\limits_{0}^{t}\exp[-\theta_n(t-\tau)]\Lambda_r(\tau;\omega)d\tau,$ (24)
where $\mu=\alpha L$ and $\theta_n=\overline{K}(n^2\pi^2+\mu^2)/(S_sL^2)$. It implies from Eqs. (3), (15) and (24) that
at the arbitrary location $x=x_\varepsilon,$
$\quad \Lambda_q(t;\omega)=\Lambda_{q_x}(x_\varepsilon,t;\omega)$
$\quad =-2\dfrac{\overline{K}}{S_sL}\sum\limits_{n=1}^{n=\infty}\dfrac{1-\cos(n\pi)}{n\pi}[n\pi\cos(n\pi\Upsilon)-\mu\sin(n\pi\Upsilon)]\int\limits_{0}^{t}\exp[-\theta_n(t-\tau)]\Lambda_r(\tau;\omega)d\tau,$ (25)
where $\Upsilon=x_\varepsilon/L$. This means that the impulse response function of the system $\varphi$ in Eqs.
(1) or (5) is taken in the form
$\quad \varphi(t,\tau)=-2\dfrac{\overline{K}}{S_sL}\sum\limits_{n=1}^{n=\infty}\dfrac{1-\cos(n\pi)}{n\pi}[\cos(n\pi\Upsilon)-\mu\sin(n\pi\Upsilon)]\exp[-\theta_n(t-\tau)].$ (26)

**4 Results and discussion**

Equation (25) implies that the transfer function $|\Lambda_q|^2$ depends on the oscillatory
function of the temporal random rainfall process; consequently, to complete the



analysis of the transfer function the oscillatory function of the temporal random
rainfall process must be specified. It is assumed that the generated temporal random
perturbations of rainfall field are governed by the noise forced diffusive rainfall model
(North et al., 1993)
$\tau_0 \dfrac{\partial}{\partial t}\rho(x,t) = \lambda_0^2 \dfrac{\partial^2}{\partial x^2}\rho(x,t) - \rho(x,t) + \xi(t)$,    (27)
where $\rho$ is a zero-mean rainfall rate perturbation, $\tau_0$ and $\lambda_0$ are the characteristic time
and length scales, respectively, which are inherent to the rainfall field, and $\xi$ is a
zero-mean random stationary forcing process which has a spectral representation of
the form (e.g., Lumley and Panofsky, 1964)
$\xi(t) = \displaystyle\int_{-\infty}^{\infty} e^{i\omega t}dZ_\xi(\omega)$.    (28)
In Eq. (27), the rainfall-rate field is represented as a first-order continuous
autoregressive process in time and an isotropic second-order autoregressive process in
space.

Furthermore, the rest of this study takes into account that rain falls within a

defined period of time over a certain area of horizontal extension from $x = -\ell$  to $x = \ell$.
As such, the initial and boundary conditions for rainfall rate perturbations may be
represented by
$\rho(x,0) = 0$,    (29a)
$\rho(-\ell,t) = 0$,    (29b)





$\rho(\ell,t) = 0$ .                                                                       (29c)

**4.1    Nonstationary random rainfall fields in time**

Using the Fourier-Stieltjes integral representation for the perturbation $\rho$,
$\rho(x,t) = \int_{-\infty}^{\infty} \Lambda_\rho(t;\omega)dZ_\xi(\omega)$,                    (30)
and Eq. (28) in Eq. (27), it follows that
$\tau_0 \dfrac{\partial}{\partial t} \Lambda_\rho(x,t;\omega) = \lambda_0^2 \dfrac{\partial^2}{\partial x^2} \Lambda_\rho(x,t;\omega) - \Lambda_\rho(x,t;\omega) + e^{i\omega t}$,    (31)
where $\Lambda_\rho$ is the oscillatory function of the rainfall rate processes. With the application
of the initial and boundary conditions,
$\Lambda_\rho(x,0;\omega) = 0$,                                                               (32a)
$\Lambda_\rho(-\ell,t;\omega) = 0$,                                                           (32b)
$\Lambda_\rho(\ell,t;\omega) = 0$,                                                            (32c)
the solution of Eqs. (31) and (32) is given by (see Appendix B)
$\Lambda_\rho(x,t;\omega) = 2\sum_{m=1}^{m=\infty} \dfrac{1-\cos(m\pi)}{m\pi} \sin(m\pi \dfrac{x+\ell}{2\ell}) \dfrac{\exp(i\Omega_t) - \exp(-\Theta_m t/\tau_0)}{\Theta_m + i\Gamma}$,    (33)
where $\Theta_m = 1 + m^2\pi^2\eta^2$, $\eta = \lambda_0/(2\ell)$, $\Omega_t = \omega t$, and $\Gamma = \omega\tau_0$.

In the case where the regional confined aquifer is directly recharged by rainfall at

the aquifer outcrop ($x = 0$), the oscillatory function is reduced to
$\Lambda_r(t;\omega) = \Lambda_\rho(0,t;\omega) = 2\sum_{m=1}^{m=\infty}\frac{1-\cos(m\pi)}{m\pi}\sin(m\frac{\pi}{2})\frac{\exp(i\Omega_t)-\exp(-\Theta_m t/\tau_0)}{\Theta_m+i\Gamma}$ .     (34)
Correspondingly, the power spectrum of rainfall rate, $S_{rr}(t,\omega)$, can be expressed by
$S_{rr}(t;\omega) = \left|\Lambda_r(t;\omega)\right|^2 S_{\xi\xi}(\omega)$
$= 4\sum_{n=1}^{n=\infty}\sum_{m=1}^{m=\infty}\frac{1-\cos(m\pi)}{m\pi}\frac{1-\cos(n\pi)}{n\pi}\sin(m\frac{\pi}{2})\sin(n\frac{\pi}{2})\frac{1}{\Theta_m^2+\Gamma^2}\frac{1}{\Theta_n^2+\Gamma^2}$
$\left\{(\Theta_m\Theta_n+\Gamma^2)[1+T_1-T_2\cos(\Omega_t)]-T_3\Gamma(\Theta_m-\Theta_n)\sin(\Omega_t)\right\}S_{\xi\xi}(\omega)$,     (35)
where $T_1 = \exp[-(\Theta_m+\Theta_n)t/\tau_0]$, $T_2 = \exp(-\Theta_m t/\tau_0)+\exp(-\Theta_n t/\tau_0)$, and $T_3 =$
$\exp(-\Theta_m t/\tau_0)-\exp(-\Theta_n t/\tau_0)$.
The transfer function of the rainfall processes in Eq. (35) behaves like a filter,
attenuating the high-frequency part of the rainfall spectrum. The graph of transfer
function, which is characterized by the characteristic time scale $\tau_0$ for different
characteristic length scales, is shown in Fig. 1. It clearly shows a reduction of the
transfer function with increasing $\tau_0$, implying a reduction of the variability of the
rainfall field with the characteristic time scale of the rainfall field. A larger $\tau_0$
decreases the temporal persistence of the rainfall fluctuations, resulting in a smaller
transfer function. It is also seen that for a fixed value of the time scale, the transfer
function of the rainfall processes tends to decrease as the length scale of the rainfall
field increases. The influence of the length scale plays a similar role as the influence
of the time scale in reducing the temporal persistence of the rainfall fluctuations and
thus the variability of the rainfall field.






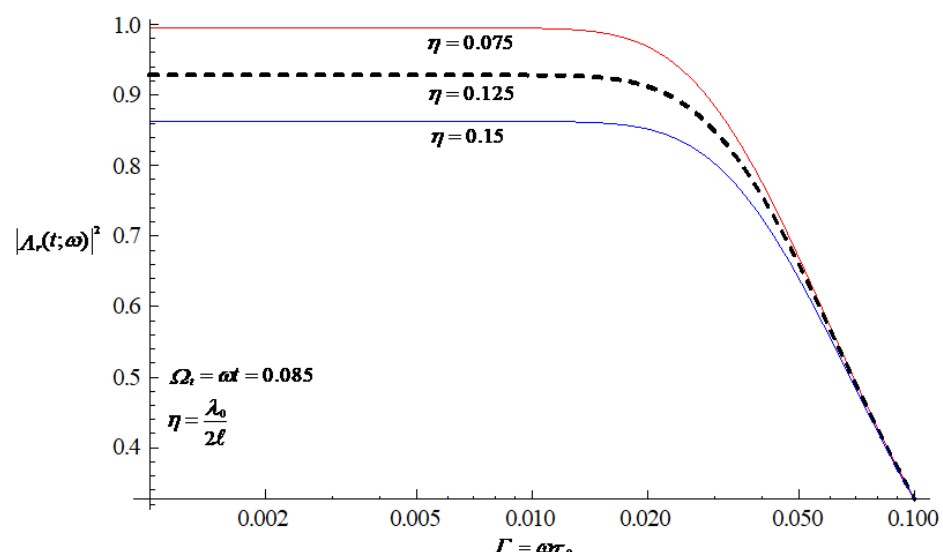


**Figure 1.** Graphical representation of the transfer function of the rainfall processes in
Eq. (35) characterized by the time scale for different length scales, where the series
calculation is truncated up to $M = N = 100$.

Through the use of Eq. (25) and Eq. (34), the oscillatory function of the
integrated discharge process could be represented as follows:
$$\Lambda_q(t;\omega) = -4\frac{\overline{K}}{S_s L}\sum_{n=1}^{n=\infty}\frac{1-\cos(n\pi)}{n\pi}\left[n\pi\cos(n\pi\Upsilon) - \mu\sin(n\pi\Upsilon)\right]$$
$$\times\sum_{m=1}^{m=\infty}\frac{1-\cos(m\pi)}{m\pi}\frac{\sin(m\frac{\pi}{2})}{\Theta_m + i\Gamma}\left[\frac{\exp(i\Omega_t) - \exp(-\theta_n t)}{\theta_n + i\omega} - \frac{\exp(-\Theta_m t/\tau_0) - \exp(-\theta_n t)}{\theta_n - \Theta_m/\tau_0}\right].$$

(36)

Thus, the transfer function of the integrated discharge flux is taken in the form



$$\frac{S_{qq}(t;\omega)}{S_{\xi\xi}(\omega)} = \left|\Lambda_q(t;\omega)\right|^2 = 16 L^2 \vartheta^2 \left\{ \left[ \sum_{n=1}^{n=\infty} \sum_{m=1}^{m=\infty} \Psi_1 \Psi_2 \left( \frac{\Theta_m \Psi_3 + \Gamma \Psi_4}{\theta_n^2 \tau_0^2 + \Gamma^2} + \frac{\Theta_m \Psi_5}{\Theta_m - \theta_n \tau_0} \right) \right]^2 \right.$$

$$\left. + \left[ \sum_{n=1}^{n=\infty} \sum_{m=1}^{m=\infty} \Psi_1 \Psi_2 \left( \frac{\Theta_m \Psi_4 - \Gamma \Psi_3}{\theta_n^2 \tau_0^2 + \Gamma^2} - \frac{\Gamma \Psi_5}{\Theta_m - \theta_n \tau_0} \right) \right]^2 \right\},$$
(37)

where $\vartheta = \overline{K}\,\tau_0/(S_s L^2)$ and
$$\Psi_1 = \frac{1}{\Theta_m^2 + \Gamma^2} \frac{1 - \cos(m\pi)}{m\pi} \sin(m\frac{\pi}{2}),$$
(38a)

$$\Psi_2 = \frac{1 - \cos(n\pi)}{n\pi} \left[ \cos(n\pi Y) - \mu \sin(n\pi Y) \right],$$
(38b)

$$\Psi_3 = \Gamma \sin(\Omega_t) + \theta_n \tau_0 \left[ \cos(\Omega_t) - \exp(-\theta_n t) \right],$$
(38c)

$$\Psi_4 = \theta_n \tau_0 \sin(\Omega_t) - \Gamma \left[ \cos(\Omega_t) - \exp(-\theta_n t) \right],$$
(38d)

$$\Psi_5 = \exp(-\Theta_m t / \tau_0) - \exp(-\theta_n t).$$
(38e)

An essential feature of the transfer function of the integrated discharge flux in Eq.
(37) is the resulting filtering associated with the flow process, as shown in Fig. 2. The
attenuating the high-frequency part of the flow discharge spectrum means that the
flow process smooths-out much of the small-scale variations caused by the rainfall
field. Physically, this feature implies that the flow field is much smoother than the
rainfall field. The figure also shows that the transfer function at fixed values for
frequency and time increases with the increasing thickness of the confined aquifer. An
increase in the thickness of the aquifer leads to an increased temporal persistence of
the flow discharge fluctuations caused by the variation of the rainfall field and thus to
an increase in the variability of integrated discharge field. As shown in Fig. 3, the
ratio of the mean hydraulic conductivity to the storage coefficient (often referred to as



the aquifer diffusivity) plays a similar role in influencing the variation of the transfer
function as the thickness of the confined aquifer. The introduction of a larger aquifer
diffusivity leads to a larger transfer function of integrated discharge and thus to a
larger variability of the discharge field. Since the variability of the discharge field is
positively correlated with that of rainfall field, the variability of the integrated
discharge field will decrease with increasing characteristic time or length scale of the
rainfall field (see Fig. 1).

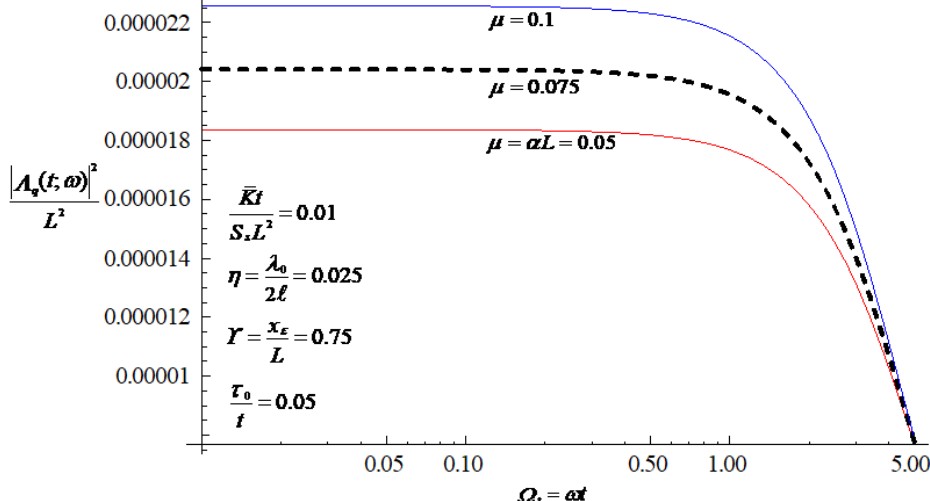


**Figure 2.** Influence of the thickness of the confined aquifer on the transfer function of
the discharge flux, where the series calculation is truncated up to $M = N = 100$.



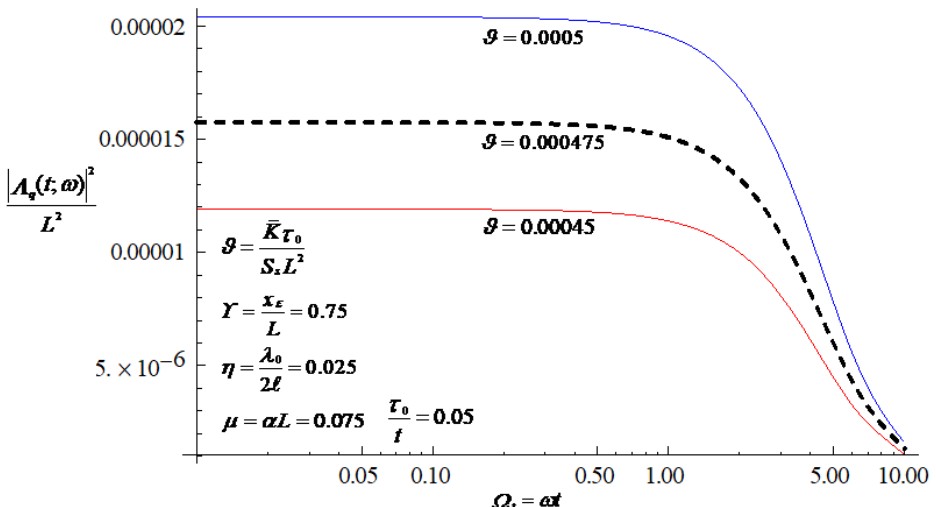


**Figure 3.** Influence of the aquifer diffusivity on the transfer function of the discharge

flux, where the series calculation is truncated up to $M = N = 100$.


From Eqs. (4) or (8), the transfer function can be defined as the ratio of the
fluctuations of an observation of output time series to those of input time series in
frequency domain. Equations (35) and (37) indicate that the transfer functions are
related to the properties of the rainfall field and the aquifer, such as the characteristic
scales of time and length of rainfall field and the diffusivity and thickness parameters
of the aquifer. Therefore, the transfer function derived here has the potential to
perform a parameter estimation based on the observations of input and output time
series using the inverse modeling approach.
Good modeling practice requires an assessment of the uncertainties associated



with the model predictions. The variance can be treated as a quantitative measure of
the uncertainty. A result such as the integration of Eq. (37) over the frequency domain
for a given spectrum of observed inflow variations could serve as a calibration target
when applying the mean value model to field situations. The mean discharge can be
determined from the mean value of Eq. (1) with the impulse response function defined
by Eq. (26).
Climate changes have a direct influence on the rainfall event (e.g., Trenberth, 2011;
Pendergrass et al., 2014; Eekhout et al., 2018). The nonstationarity in the statistical
properties of rainfall field is a representation of climate change (e.g., Razavi et al.,
2015; López and Francés, 2013; Benoit et al., 2020). The effect of climatic change on
variability of groundwater specific discharge has not yet been well characterized in
the literature. The transfer function in Eq. (37), which relates the nonstationary
spectra of the rainfall fluctuations to those of integrated discharge variation, has the
potential to analyze the effects of climate change on groundwater specific discharge
variability.

**4.2   A note on stationary random rainfall fields in time**

If the temporal random rainfall fields are stationary, there exists a representation of



the rainfall perturbation process in terms of a Fourier-Stieltjes integral as Eq. (6).
Substituting Eqs. (6) and (21) into Eq. (19) gives
$\dfrac{S_s}{\overline{K}}\dfrac{\partial}{\partial t}\Lambda_h(x,t;\omega) = \dfrac{\partial^2}{\partial x^2}\Lambda_h(x,t;\omega) + 2\alpha\dfrac{\partial}{\partial x}\Lambda_h(x,t;\omega) + \dfrac{e^{-\alpha x}}{\beta\overline{K}}e^{i\omega t}$ .            (39)
The solution of Eq. (39) with conditions Eq. (23) is
$\Lambda_h(x,t;\omega) = \dfrac{2}{S_s\beta}\sum\limits_{n=1}^{n=\infty}\dfrac{1-\cos(n\pi)}{n\pi}\exp(-\mu\dfrac{x}{L})\sin(n\pi\dfrac{x}{L})\dfrac{\exp(i\Omega_t)-\exp(-\theta_n t)}{\theta_n+i\omega}$ ,            (40)
so that
$\Lambda_q(t;\omega) = -2\dfrac{\overline{K}}{S_s L}\sum\limits_{n=1}^{n=\infty}\dfrac{1-\cos(n\pi)}{n\pi}\left[n\pi\cos(n\pi\Upsilon)-\mu\sin(n\pi\Upsilon)\right]\dfrac{\exp(i\Omega_t)-\exp(-\theta_n t)}{\theta_n+i\omega}$ .            (41)
and thus
$\left|\Lambda_q(t;\omega)\right|^2 = 4L^2\mathcal{G}^2\sum\limits_{n=1}^{n=\infty}\sum\limits_{m=1}^{m=\infty}\dfrac{\Phi(m)\Phi(n)}{(\theta_n^2\tau_0^2+\Gamma^2)(\theta_m^2\tau_0^2+\Gamma^2)}\left[(\theta_m\theta_n\tau_0^2+\Gamma^2)(1+\Delta_1-\cos(\Omega_t)\Delta_2)-\Gamma\sin(\Omega_t)(\theta_m-\theta_n)\tau_0\Delta_3\right]$

,

(42)

where
$\Phi(y) = \dfrac{1-\cos(y\pi)}{y\pi}\left[y\pi\cos(y\pi\Upsilon)-\mu\sin(y\pi\Upsilon)\right]$ ,            (43)
$\Delta_1 = \exp[-(\theta_m+\theta_n)t]$, $\Delta_2 = \exp(-\theta_m t)+\exp(-\theta_n t)$, and $\Delta_3 = \exp(-\theta_m t)-\exp(-\theta_n t)$.

At large times, Eq. (35) approach a finite value as

$S_{rr}(\omega) = 4\sum\limits_{n=1}^{n=\infty}\sum\limits_{m=1}^{m=\infty}\dfrac{1-\cos(m\pi)}{m\pi}\dfrac{1-\cos(n\pi)}{n\pi}\sin(m\dfrac{\pi}{2})\sin(n\dfrac{\pi}{2})\dfrac{\Theta_m\Theta_n+\Gamma^2}{(\Theta_m^2+\Gamma^2)(\Theta_n^2+\Gamma^2)}S_{\xi\xi}(\omega)$ .            (44)
and the corresponding rainfall process is stationary. Combining Eq. (42) with Eq. (44)
gives
$\dfrac{S_{qq}(\omega)}{S_{\xi\xi}(\omega)} = 16L^2\mathcal{G}^2\left\{\sum\limits_{n=1}^{n=\infty}\sum\limits_{m=1}^{m=\infty}\dfrac{\Phi(m)\Phi(n)}{(\theta_n^2\tau_0^2+\Gamma^2)(\theta_m^2\tau_0^2+\Gamma^2)}\left[(\theta_m\theta_n\tau_0^2+\Gamma^2)(1+\Delta_1-\cos(\Omega_t)\Delta_2)-\Gamma\sin(\Omega_t)(\theta_m-\theta_n)\tau_0\Delta_3\right]\right\}$




$$\times\left[\sum_{n=1}^{n=\infty}\sum_{m=1}^{m=\infty}\frac{1-\cos(m\pi)}{m\pi}\frac{1-\cos(n\pi)}{n\pi}\sin(m\frac{\pi}{2})\sin(n\frac{\pi}{2})\frac{\Theta_m\Theta_n+\Gamma^2}{(\Theta_m^2+\Gamma^2)(\Theta_n^2+\Gamma^2)}\right]. \qquad (45)$$

Note that the nonstationarity in the hydraulic head or integrated discharge is
introduced by a nonuniform thickness of the confined aquifer, even if the recharge
field is stationary. Nonuniformity in the mean flow, for example, can also cause the
nonstationarity in the statistics of random flow fields in heterogeneous aquifers (e.g.,
Rubin and Bellin, 1994; Ni and Li, 2006; Ni et al., 2010).

**5   Conclusions**

An analytical transfer function is developed to describe the spectral response
characteristics of confined aquifers with variable thickness to the variation of the
rainfall field, where the aquifer is directly recharged by rainfall at the outcrop of the
aquifer. The rainfall-discharge process is treated as nonstationary in time, as it reflects
the stochastic nature of the hydrological process. Any varying rainfall input at any
time resolution can be convolved with the transfer function (or impulse response
function) to simulate any discharge output of a linear model. The transfer function
derived here, which relates the nonstationary spectra of the rainfall fluctuations to
those of integrated discharge variation, has the potential to analyze the influence of





climate change on groundwater recharge variability.
The closed-form results of this work are developed on the basis of the
Fourier-Stieltjes representation approach, which allows to analyze the effects of the
controlling parameters in the models on the transfer function of the integrated
discharge. It is founded that the persistence of rainfall fluctuations is greater for a
smaller value of the characteristic time or length scale of the rainfall field, which in
turn leads to greater variability of the integrated discharge field. The attenuating
characteristic of the confined aquifer flow system is observed in the spectral domain.
The variability of the integrated discharge in confined aquifer with variable thickness
is increased with the thickness parameter $\alpha$. The larger the aquifer diffusivity, the
greater the spectrum (variability) of the integrated discharge.

**Appendix A:    Evaluation of $\Lambda_h$ in Eq. (20)**

The boundary-value problem describing the depth-averaged head fluctuations induced
by the variation of recharge rate in frequency domain is given by Eqs. (22) and (23).
Using the transformation,
$\Lambda_h(x,t;\omega) = \exp[-\alpha(x + \frac{\alpha \overline{K}}{S_s}t)]U(x,t;\omega)$,                    (A1)
Eq. (22) in $\Lambda_h(x,t;\omega)$ together with Eq. (23) can be converted into a new (easier) one





in a new variable $U(x,t;\omega)$ as
$\dfrac{\partial}{\partial t}U(x,t;\omega) = \dfrac{\overline{K}}{S_s}\dfrac{\partial^2}{\partial x^2}U(x,t;\omega) + \dfrac{1}{\beta S_s}\exp(\dfrac{\overline{K}\alpha^2}{S_s}t)\varLambda_r(t;\omega)$,    (A2)
with
$U(x,0;\omega) = 0$,    (A3a)
$U(0,t;\omega) = 0$,    (A3b)
$U(L,t;\omega) = 0$.    (A3c)
The solution of Eqs. (A2) and (A3) can be found by the technique of separation of
variables (e.g., Farlow, 1993) as
$U(x,t;\omega) = \dfrac{2}{S_s\beta}\displaystyle\sum_{n=1}^{n=\infty}\dfrac{1-\cos(n\pi)}{n\pi}\sin(n\pi\dfrac{x}{L})\int_0^t \exp[-\upsilon_n(t-\tau)]\exp(\dfrac{\overline{K}}{S_s}\alpha^2\tau)\varLambda_r(\tau;\omega)d\tau$,    (A4)
where $\upsilon_n = \overline{K}n^2\pi^2/(S_sL^2)$. With reference to Eq. (A1), the solution of Eqs. (22) and (23)
is then given by Eq. (24).

**Appendix B:   Evaluation of $\varLambda_\rho$ in Eq. (31)**

Making use of the transformation,
$\varLambda_\rho(x,t;\omega) = \exp(-\dfrac{t}{\tau_0})u(x,t;\omega)$,    (B1)
leads Eqs. (31) and (32) to
$\dfrac{\partial}{\partial t}u(x,t;\omega) = \dfrac{\lambda_0^2}{\tau_0}\dfrac{\partial^2}{\partial x^2}u(x,t;\omega) + \dfrac{1}{\tau_0}\exp[(\dfrac{1}{\tau_0}+i\omega)t]$,    (B2)
with

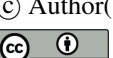



$u(x,0;\omega) = 0$,    (B3a)
$u(-\ell,t;\omega) = 0$,    (B3b)
$u(\ell,t;\omega) = 0$.    (B3c)
In a similar way, based on the technique of separation of variables, Eqs. (B2) and (B3)
arrive at the solution in the form
$$u(x,t;\omega) = 2\sum_{m=1}^{m=\infty}\frac{1-\cos(m\pi)}{m\pi}\sin(m\pi\frac{x+\ell}{2\ell})\frac{\exp[(1+i\Gamma)t/\tau_0]-\exp(-\varsigma_m t/\tau_0)}{\Theta_m+i\Gamma},$$    (B4)
where $\varsigma_m = m^2\pi^2\eta^2$, $\eta = \lambda_0/(2\ell)$, $\Theta_m = 1+\varsigma_m$, and $\Gamma = \omega\tau_0$. The use of Eqs. (B1) and (B4)
results in Eq. (33).

*Data availability*. No data was used for the research described in the article.

*Author contributions*. C-MC: Conceptualization, Methodology, Formal analysis,
Writing - original draft preparation, Writing - review & editing.
C-FN: Conceptualization, Methodology, Formal analysis, Writing - original draft
preparation, Writing - review & editing, Supervision, Funding acquisition.
W-CL: Conceptualization, Methodology, Formal analysis, Writing - original draft
preparation, Writing - review & editing.
C-PL: Conceptualization, Methodology, Formal analysis, Writing - original draft
preparation, Writing - review & editing.
I-HL: Conceptualization, Methodology, Formal analysis, Writing - original draft
preparation, Writing - review & editing.



*Competing interests.* The authors declare that they have no conflict of interest.

*Acknowledgements.* Research leading to this paper has been partially supported by
the grant from the Taiwan Ministry of Science and Technology under the grants
MOST 108-2638-E-008-001-MY2, MOST 108-2625-M-008 -007, and MOST
107-2116-M-008 -003 -MY2.

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



**Figure captions**

**Figure 1.** Graphical representation of the transfer function of the rainfall processes in Eq. (35) characterized by the time scale for different length scales, where the series calculation is truncated up to $M = N = 100$.

**Figure 2.** Influence of the thickness of the confined aquifer on the transfer function of the discharge flux, where the series calculation is truncated up to $M = N = 100$.

**Figure 3.** Influence of the aquifer diffusivity on the transfer function of the discharge flux, where the series calculation is truncated up to $M = N = 100$.