# Peer review of "Technical note: Discharge response of a confined aquifer with variable thickness to temporal nonstationary random recharge processes"

_Hydrology and Earth System Sciences, 2020_

## Referee Comment (RC1) · Anonymous Referee #1 · 20 Feb 2021

Review of DISCHARGE RESPONSE OF A CONFINED AQUIFER WITH VARIABLE THICKNESS TO TEMPORAL NONSTATIONARY RANDOM RECHARGE PROCESSES

This paper provides an analysis, through a lumped model, of confined aquifer discharge to random recharge. It also accounts for variable thickness of the confined system. The authors propose a closed-form expression for the transfer function which will be employed in the linear lumped confined aquifer model.

Although interesting solutions are developed from this study (therefore there is a technical merit to this work), I have concerns regarding the following points:

1. Justification for the model adopted. It neglect heterogeneity and the 1D approximation is not well justified for this reviewer given the knowledge of the works done in the past. The model presented by the authors does not account for heterogeneity and is a 1D system. It also averages out variability along $x_3$.

2. The authors neglect a large body of literature that worked on this topic of aquifer recharge and flow (and even transport where these effects are more pronounced). There are a series of works published where authors looked at aquifer recharge (stochastic and deterministic) in the presence of heterogeneity in 2d and 3d flows. Furthermore, there is also a body of recent literature looking at the uncertainty related to climate projection on aquifer flow and transport. Please see my detailed comments below.

3. The applicability of this model. How does this model improve our fundamental knowledge with respect to the previously published works that tackled more complex flow conditions? This needs to be addressed.

Therefore, I believe this technical note needs to be revised prior to its exposure. Below I provide more details on my comments that range from minor to major.

- Please revise equation notations. For example, in eqn 10, the square brackets are small and do not match the size of the fraction (i.e. partial h partial x).

- Line 129: I do not agree with the authors when they write "small fluctuations in the $x_1$ and $x_2$ directions" led a 1D flow. Many authors in the stochastic hydrogeological community have shown that variability in $x_1$ and $x_2$ can impact meandering of the streamlines – even for log-conductivity variances smaller than one.

- In fact, the work under revision neglects a few contributions on the presence of aquifer recharge in heterogeneous aquifers. For example, see works by:

o Rubin, Y., & Bellin, A. (1994). The effects of recharge on flow nonuniformity and macrodispersion. Water resources research, 30(4), 939-948. o Li, L., & Graham, W. D. (1998). Stochastic analysis of solute transport in heterogeneous aquifers subject to spatially random recharge. Journal of hydrology, 206(1-2), 16-38. o Ciriello, V., & de Barros, F. P. J. (2020). Characterizing the Influence of Multiple Uncertainties on Predictions of Contaminant Discharge in Groundwater Within a Lagrangian Stochastic Formulation. Water Resources Research, 56(10), e2020WR027867. o Destouni, G., Simic, E., & Graham, W. (2001). On the applicability of analytical methods for estimating solute travel time statistics in nonuniform groundwater flow. Water Resources Research, 37(9), 2303-2308. - Later the authors talk about the lack of studies looking at climate change and its impact in the subsurface environment. See for example the work of Libera et al (2019) where it was shown how changes in recharge impact flow rates and consequentially transport and aquifer remediation strategies. o Libera, A. et al. (2019). Climate change impact on residual contaminants under sustainable remediation. Journal of Contaminant Hydrology, 226, 103518.

- What is the justification for expression 18? Why an exponentially variable thickness? Is this used for a mathematical convenience? Please justify.

- Applicability of this model: Are there any data to compare the results of this model to? This might be out of the scope of this work but it would be nice to see additional discussion related to the applicability of this model and its range of validity.

- Please highlight the key novelties of this work with respect to the previously published literature on recharge and aquifers.

---

## Referee Comment (RC2) · Anonymous Referee #2 · 25 Feb 2021

The manuscript provides a closed-form expression for the transfer function in the frequency domain of a confined aquifer with variable thickness submitted to a variable recharge rate. The mathematical development is well described and quite easy to follow. I suggest intermediate revisions.

Here are some general comments: 1) I believe that a conceptual scheme of the system including the main variables of the problem would helpfully accompany the mathematical development 2) The assumptions should be discussed to better assess the capabilities and limitations of the solution proposed: - homogeneity: what kind of sys-

tems (possibly known aquifers) would be fairly well modelled by the solution proposed? To what extent? - one-dimensional flow: I expect strong limitations of this assumption given the thickness variability and possible convergence or divergence of flow One would obviously not expect a homogeneous and 1D solution to represent the complexity of natural systems. However, it can still be used as a practical rough approximation. 3) The sensitivity analysis going along with figures 1, 2 and 3 would greatly benefit from a mechanistic interpretation and/or (a short) comparison with existing works. It would reinforce the validity and usefulness of the solution proposed. This needs more effort.

Some specific comments: - There are no boundary conditions nor initial conditions for equation 10 - Line 91: may serve as a basis? not "service" - Line 119: I am not sure that b1 and b2 are defined - Line 149: "increases", not increase - Line 149: be more specific: "increase exponentially" (as shown in eq 18)

—————————————————

---

## Author Comment (AC1) · 28 Mar 2021

**Response to comments of Anonymous Referee 2**

We would like to thank the referee for the valuable comments and suggestions, which improved the quality of the paper. Below you will find our response in regard to the comments and suggestions.

**Comments to the Authors:**
The manuscript provides a closed-form expression for the transfer function in the frequency domain of a confined aquifer with variable thickness submitted to a variable recharge rate. The mathematical development is well described and quite easy to follow.
I suggest intermediate revisions.

Here are some general comments:
1) I believe that a conceptual scheme of the system including the main variables of the problem would helpfully accompany the mathematical development.
**Response**
    To make the focus of this study clear, we have changed the first paragraph in Section 2, "Problem Statement," to read as follows

    "In certain areas, aquifer recharge can vary greatly over time, so determining the discharge of the aquifer at the outlet for regional groundwater problems, which involves transferring recharge at the aquifer outcrop over a relatively large space scale, can be quite difficult. However, it is very important for planning and management of regional groundwater resources that require knowledge of discharge at the aquifer outlet over a long period of time. This study is therefore devoted to quantifying the discharge response of the confined aquifer at the outlet to the temporal variation in aquifer recharge.

    In this study, a confined aquifer with variable thickness is considered as a linear block-box system with a stochastic rainfall recharge input and therefore a stochastic runoff output. Both inputs and outputs are variable in time. In a linear system, the output of the system can be represented as a linear combination of the responses to each of the basic inputs through the convolution integral on a continuous time scale as (e.g., Rugh, 1981; Rinaldo and Marani, 1987)

$$Q(t) = \int_0^t \varphi(t, \tau) R(\tau) d\tau, \tag{1}$$

where $Q$ and $R$ denote the output flow (discharge) rate and the input flow

(recharge) rate of the system, respectively, and $\varphi$ is the impulse response function of the system. As shown in Fig. 1, once an appropriate impulse response function can be specified at the scale of the aquifer, it is possible to evaluate the system response from records of the input without the need to specify smaller scale heterogeneity. As will be shown below, the transfer function of the system can be used to characterize the uncertainty (variability) expected in applying the convolution integral Eq. (1) to the regional groundwater flow problems.

[Figure]

**Figure 1.** Schematic representation of a linear block-box system."

2) The assumptions should be discussed to better assess the capabilities and limitations of the solution proposed: - Homogeneity: what kind of systems (possibly known aquifers) would be fairly well modelled by the solution proposed? To what extent? - One-dimensional flow: I expect strong limitations of this assumption given the thickness variability and possible convergence or divergence of flow. One would obviously not expect a homogeneous and 1D solution to represent the complexity of natural systems. However, it can still be used as a practical rough approximation.

**Response**

    a. In regional aquifers, the relatively small depth (compared to the horizontal dimensions) allows modelers to simplify the 3-D flow problem in a 2-D problem. Under such a condition, the 2-D flow equation (please see Eq. (11)) is governed by the transmissivity (defined as the hydraulic conductivity times the depth) and depth-averaged hydraulic head (please see Eq. (12)). Transmissivity and depth-averaged head account for variability in both thickness and conductivity in the $x_3$ direction. Furthermore, the assumption of unidirectional mean regional groundwater flow in the $x_1$ direction allows the 2-D flow equation to be simplified to a 1-D flow equation. The approximation of 1-D flow equation is simply due to the fact that the flow domain in the $x_1$ direction is much larger than that in the $x_2$ direction. The transmissivity and

depth-averaged head appearing in 1-D flow equation still account for variability in the $x_1$ and $x_3$ directions.

b. "The use of the depth-averaged hydraulic head operator for modeling regional groundwater flow is valid when the variation in aquifer thickness is much smaller than the average thickness (Bear, 1979; Bear and Cheng, 2010). The error introduced by the use of this operator is very small in most cases of practical interest, greatly simplifying the analysis of flow in confined aquifers."

The above sentences are added on the page 9 (Line 144).

c. "It is worth noting that a one-dimensional flow equation with the transmissivity parameter has been widely used to predict the regional groundwater flow fields in the downstream region of the aquifer in field applications (e.g., Gelhar, 1974; Onder, 1998; Molénat et al., 1999; Russian et al., 2013)."

The above note is added on page 10 (Line160).

Gelhar, L.: Stochastic analysis of phreatic aquifers, Water Resour. Res., 10(3), 539-545, 1974.

Molénat, J., Davy, P., Gascuel-Odoux, C., and Durand, P.: Study of three subsurface hydrologic systems based on spectral and cross-spectral analysis of time series, J. Hydrol., 222(1-4), 152-164, 1999.

Onder, H.: One-dimensional transient flow in a finite fractured aquifer system, Hydrol. Sci. J., 43(2), 243-265, 1998.

Russian, A., Dentz, M., Le Borgne, T., Carrera, J., and Jimenez-Martinez, J.: Temporal scaling of groundwater discharge in dual and multicontinuum catchment models, Water Resour. Res., 49(12), 8552-8564, 2013.

3) The sensitivity analysis going along with figures 1, 2 and 3 would greatly benefit from a mechanistic interpretation and/or (a short) comparison with existing works. It would reinforce the validity and usefulness of the solution proposed. This needs more effort.

**Response**

a. To our knowledge, modeling of the natural recharge-discharge process in confined aquifers of nonuniform thickness as a nonstationary process has not been previously presented in the literature. No data are available to compare the results of this work. An application of the proposed model to predict the outflow discharge is added on page 22 (Line 361) as

"**4.2  Application in the prediction of outflow discharge**

The usefulness of the stochastic theory presented here lies in its essentially predictive nature. The variance can be used as a quantification of the uncertainty associated with the prediction in field situations using the linear system model. In this sense, the solution of Eq. (1) ± two times the square root of the variance provides a rational framework for predicting discharge over a relatively large spatial scale where direct observations of such a dependent variable are not possible.

For large times, the first term in Eq. (37) dominates the sum of the other terms, and therefore the transfer function can be approximated by

$$\left|\Lambda_q(t;\omega)\right|^2 = \frac{256}{\pi^2}L^2\mathcal{G}^2\left[\pi\cos(\pi\Upsilon)-\mu\sin(\pi\Upsilon)\right]^2\frac{1}{\Theta_1^2+\Gamma^2}\frac{1}{\mathbb{V}^2+\Gamma^2}$$

$$\left[1+2\mathbb{V}\Xi\,T_s+T_s^2-2(\mathbb{V}\Xi+T_s)\cos(\Omega_t)-2\Xi\,\Omega_t\sin(\Omega_t)+\Xi^2\right], \tag{42}$$

where $\Theta_l = 1 + \pi^2\eta^2$, $\mathbb{V} = \overline{K}\,\tau_0(\pi^2+\mu^2)/(S_sL^2)$, $\Xi = \left[\exp(-\Theta_l t/\tau_0)-T_s\right]/(\mathbb{V}-\Theta_l)$, and $T_s = \exp(-\mathbb{V}t/\tau_0)$. If the variation of the rainfall event is generated by a random white noise forcing, the variance of the outflow discharge at large times can then be calculated using Eq. (42) as

$$\sigma_q^2(t) = \int_{-\infty}^{\infty}S_{qq}(t;\omega)d\omega = \int_{-\infty}^{\infty}\left|\Lambda_q(t;\omega)\right|^2 S_{\xi\xi}(\omega)d\omega$$

$$= \frac{256}{\pi^2}L^2\mathcal{G}^2G_0\left[\pi\cos(\pi\Upsilon)-\mu\sin(\pi\Upsilon)\right]^2\left\{\frac{\Xi^2}{\Theta_1}+\frac{1+2\mathbb{V}\Xi\,T_s+T_s^2}{\mathbb{V}\Theta_1(\mathbb{V}+\Theta_1)}\right.$$

$$\left.-2(\mathbb{V}\Xi+T_s)\frac{\Theta_1\cosh(\mathbb{V}t/\tau_0)-\mathbb{V}\cosh(\Theta_l t/\tau_0)}{\mathbb{V}\Theta_1^3+\Theta_1\mathbb{V}^3}-2\Xi\frac{\sinh(\Theta_l t/\tau_0)-\sinh(\mathbb{V}t/\tau_0)}{\Theta_1^2-\mathbb{V}^2}\right\}, \tag{43}$$

where $G_0$ represents a constant spectral density of a white noise process. Note that white noise is a signal that contains all frequencies in equal proportions, that is, a signal whose spectrum is flat.

After observing the recharge rate $R(t)$ over time at the outcrop of the aquifer and identifying input parameters such as the specific storage coefficient, mean hydraulic conductivity and geometrical parameters of the aquifer and the characteristic time and length scales of the rainfall event for a given area or region, the discharge can be determined under uncertainty in the far downstream aquifer area, Eq. (1) together with Eq. (26) ± two times the square root of Eq. (43). It provides an important basis for the rational management of regional groundwater resources in complex geologic settings under uncertainty."

b. A note on the validity of the proposed model is added on page 17 (Line 290) as

"Note that the linearity in modeling the recharge-discharge response of a catchment in Eq. (1), which was originally developed for large catchments, increases with catchment area (e.g., Chow et al., 1988). This implies that the impulse responses and transfer functions derived here are valid in large confined aquifers."

Chow, V. T., Maidment, D. R., and Mays, L. W.: Applied hydrology, McGraw-Hill, 1988.

Some specific comments:

- There are no boundary conditions nor initial conditions for equation 10.

**Response**

In the development of Eq. (11) from Eq. (10), only the boundary conditions are used, namely the slip-free condition, at a fixed boundary the fluid has zero velocity. To clarify that, we add a note on page 9 (Line 141) as

"All terms involved in the fluxes in the directions of $x_1$ and $x_2$ at the boundaries are removed due to the no-slip condition at the boundaries."

- Line 91: may serve as a basis? not "service"

**Response**

The typo has been corrected.

- Line 119: I am not sure that $b_1$ and $b_2$ are defined

**Response**

The definitions of $b_1$ and $b_2$ were added on page 9 (Line L132) as

"$b_1(x_1,x_2)$ and $b_2(x_1,x_2)$ are the elevations of the fixed bottom and ceiling of the confined aquifer, respectively,"

- Line 149: "increases", not increase

**Response**

The typo has been corrected.

- Line 149: be more specific: "increase exponentially" (as shown in Eq.(18)).

**Response**

As suggested, it has been changed to "increases exponentially".

---

## Author Comment (AC2) · 28 Mar 2021

**Response to comments of Anonymous Referee 1**

We would like to thank the referee for the valuable comments and suggestions, which improved the quality of the paper. Below you will find our response in regard to the comments and suggestions.

**Comments to the Authors:**

This paper provides an analysis, through a lumped model, of confined aquifer discharge to random recharge. It also accounts for variable thickness of the confined system. The authors propose a closed-form expression for the transfer function which will be employed in the linear lumped confined aquifer model.

Although interesting solutions are developed from this study (therefore there is a technical merit to this work), I have concerns regarding the following points:

1. Justification for the model adopted. It neglect heterogeneity and the 1D approximation is not well justified for this reviewer given the knowledge of the works done in the past. The model presented by the authors does not account for heterogeneity and is a 1D system. It also averages out variability along $x_3$.

**Response**

The proposed stochastic model presents itself as one-dimensional, but it takes into account variability in the directions of $x_1$ and $x_3$.

a. In regional aquifers, the relatively small depth (compared to the horizontal dimensions) allows modelers to simplify the 3-D flow problem in a 2-D problem. Under such a condition, the 2-D flow equation (please see Eq. (11)) is governed by the transmissivity (defined as the hydraulic conductivity times the depth) and depth-averaged hydraulic head (please see Eq. (12)). Transmissivity and depth-averaged head account for variability in both thickness and conductivity in the $x_3$ direction. Furthermore, the assumption of unidirectional mean regional groundwater flow in the $x_1$ direction allows the 2-D flow equation to be simplified to a 1-D flow equation. The approximation of 1-D flow equation is simply due to the fact that the flow domain in the $x_1$ direction is much larger than that in the $x_2$ direction. The transmissivity and depth-averaged head appearing in 1-D flow equation still account for variability in the $x_1$ and $x_3$ directions.

b. "The use of the depth-averaged hydraulic head operator for modeling regional groundwater flow is valid when the variation in aquifer thickness is much smaller than the average thickness (Bear, 1979; Bear and Cheng, 2010). The error introduced by the use of this operator is very small in most

cases of practical interest, greatly simplifying the analysis of flow in confined aquifers."

In addition, "the depth-averaged head representation used in Eq. (41) is consistent with what is observed in the fields." Please refer to **Response G** for details.

The above sentences are added on the page 9 (Line 144).

c. "It is worth noting that a one-dimensional flow equation with the transmissivity parameter has been widely used to predict the regional groundwater flow fields in the downstream region of the aquifer in field applications (e.g., Gelhar, 1974; Onder, 1998; Molénat et al., 1999; Russian et al., 2013)."

The above note is added on page 10 (Line 160).

Gelhar, L.: Stochastic analysis of phreatic aquifers, Water Resour. Res., 10(3), 539-545, 1974.

Molénat, J., Davy, P., Gascuel-Odoux, C., and Durand, P.: Study of three subsurface hydrologic systems based on spectral and cross-spectral analysis of time series, J. Hydrol., 222(1-4), 152-164, 1999.

Onder, H.: One-dimensional transient flow in a finite fractured aquifer system, Hydrol. Sci. J., 43(2), 243-265, 1998.

Russian, A., Dentz, M., Le Borgne, T., Carrera, J., and Jimenez-Martinez, J.: Temporal scaling of groundwater discharge in dual and multicontinuum catchment models, Water Resour. Res., 49(12), 8552-8564, 2013.

2. The authors neglect a large body of literature that worked on this topic of aquifer recharge and flow (and even transport where these effects are more pronounced). There are a series of works published where authors looked at aquifer recharge (stochastic and deterministic) in the presence of heterogeneity in 2d and 3d flows. Furthermore, there is also a body of recent literature looking at the uncertainty related to climate projection on aquifer flow and transport. Please see my detailed comments below.

**Response**

For reasons why they are not cited in the manuscript, please refer to **Response C and D** for details.

3. The applicability of this model. How does this model improve our fundamental knowledge with respect to the previously published works that tackled more complex flow conditions? This needs to be addressed.

**Response**

a. The applicability of the proposed model is addressed by applying the transfer function to the field. Please refer to **Response F** for details.

b. For details on model improvements of previously published works, please refer to **Response G**.

Therefore, I believe this technical note needs to be revised prior to its exposure. Below I provide more details on my comments that range from minor to major.

**A.** Please revise equation notations. For example, in Eq. (10), the square brackets are small and do not match the size of the fraction (i.e. partial h partial x).

**Response**

As suggested, the square brackets in Eqs. (10) and (11) are adjusted to match the size of the fraction.

**B.** Line 129: I do not agree with the authors when they write "small fluctuations in the $x_1$ and $x_2$ directions" led a 1D flow. Many authors in the stochastic hydrogeological community have shown that variability in $x_1$ and $x_2$ can impact meandering of the streamlines - even for log-conductivity variances smaller than one.

**Response**

Yes.

a. To avoid confusion, we modify the first paragraph in Section 2, "Problem Statement," to clarify the focus of this study, which is to quantify the discharge response of the stressed aquifer at the outlet to temporal variation in groundwater recharge, as follows

"In certain areas, aquifer recharge can vary greatly over time, so determining the discharge of the aquifer at the outlet for regional groundwater problems, which involves transferring recharge at the aquifer outcrop over a relatively large space scale, can be quite difficult. However, it is very important for planning and management of regional groundwater resources that require knowledge of discharge at the aquifer outlet over a long period of time. This study is therefore devoted to quantifying the discharge response of the confined aquifer at the outlet to the temporal variation in aquifer recharge.

In this study, a confined aquifer with variable thickness is considered as a linear block-box system with a stochastic rainfall recharge input and therefore a stochastic runoff output. Both inputs and outputs are variable in time. In a linear system, the output of the system can be represented as

a linear combination of the responses to each of the basic inputs through the convolution integral on a continuous time scale as (e.g., Rugh, 1981; Rinaldo and Marani, 1987)

$$Q(t) = \int_0^t \varphi(t,\tau)R(\tau)d\tau \,, \tag{1}$$

where $Q$ and $R$ denote the output flow (discharge) rate and the input flow (recharge) rate of the system, respectively, and $\varphi$ is the impulse response function of the system. As shown in Fig. 1, once an appropriate impulse response function can be specified at the scale of the aquifer, it is possible to evaluate the system response from records of the input without the need to specify smaller scale heterogeneity. As will be shown below, the transfer function of the system can be used to characterize the uncertainty (variability) expected in applying the convolution integral Eq. (1) to the regional groundwater flow problems.

[Figure]

**Figure 1.** Schematic representation of a linear block-box system."

b. We change that sentence to

"In this study, the regional confined aquifer is considered with a nonuniform, unidirectional mean flow in the $x_1$-axis direction, but with small flow variations in the $x_1$- and $x_2$-axis directions and time-varying recharge at the aquifer outcrop ($x_1 = 0$). Since the regional flow domain considered in the $x_1$ direction is much larger than that in the $x_2$ direction, Eqs. (11) and (13) can be approximated as one-dimensional by…"

c. In addition, the following sentence is added immediately after that, as

"It is worth noting that a one-dimensional flow equation with the transmissivity parameter has been widely used to predict the regional groundwater flow fields in the downstream region of the aquifer in field applications (e.g., Gelhar, 1974; Onder, 1998; Molénat et al., 1999; Russian

et al., 2013)."

**C.** In fact, the work under revision neglects a few contributions on the presence of aquifer recharge in heterogeneous aquifers. For example, see works by:

Rubin, Y., & Bellin, A. (1994). The effects of recharge on flow nonuniformity and macrodispersion. Water resources research, 30(4), 939-948.

Li, L., & Graham, W. D. (1998). Stochastic analysis of solute transport in heterogeneous aquifers subject to spatially random recharge. Journal of hydrology, 206(1-2), 16-38.

Ciriello, V., & de Barros, F. P. J. (2020). Characterizing the influence of multiple uncertainties on predictions of contaminant discharge in groundwater within a Lagrangian stochastic formulation. Water Resources Research, 56(10), e2020WR027867.

Destouni, G., Simic, E., & Graham, W. (2001). On the applicability of analytical methods for estimating solute travel time statistics in nonuniform groundwater flow. Water Resources Research, 37(9), 2303-2308.

**Response**
a. Since the focus of the above references is placed on analyzing the effect of recharge on the solute transport, they all assume that the flow field is steady-state to simplify the problem. However, in this study, a transient flow field is treated.
b. Moreover, Li and Graham (1998) consider recharge as a spatial stationary random variable, while the rest of the above references consider recharge as deterministic and spatially uniformly distributed. However, this study considers recharge as a temporal random variable.
c. In addition, the manuscript takes into account that the recharge process is nonstationary in time, which has not been presented so far in the literature.

For these reasons, they are not cited in the manuscript.

**D.** Later the authors talk about the lack of studies looking at climate change and its impact in the subsurface environment. See for example the work of Libera et al (2019) where it was shown how changes in recharge impact flow rates and consequentially transport and aquifer remediation strategies.

Libera, A., de Barros, F. P. J., Faybishenko, B., Eddy-Dilek, C., Denham, M., Lipnikov, K., … Wainwright, H. (2019). Climate change impact on residual contaminants under sustainable remediation. Journal of Contaminant Hydrology, 226, 103518.

**Response**

Thanks for the reference. However, the numerical simulations of Libera et al. (2019) do not model the rainfall-recharge process as a temporally **nonstationary** process. The manuscript mentions that the climate is nonstationary in time, which in turn leads to nonstationarities in aquifer recharge. Therefore, to avoid confusion, we change that to

"The nonstationarity in the statistical properties of rainfall field is a representation of climate change (e.g., Razavi et al., 2015; López and Francés, 2013; Benoit et al., 2020). The nonstationary effect of climatic changes over time on the variability of specific groundwater discharge has not been well characterized in the literature."

**E.** What is the justification for expression 18? Why an exponentially variable thickness?

Is this used for a mathematical convenience? Please justify.

**Response**

a. Nonuniformities in the thickness of the aquifer are frequently reported. The traditional diffusion equation for regional groundwater flow through confined aquifers is originally designed for the analysis of flow field in an aquifer of uniform thickness, therefore it is very difficult to use it to assess the influence of nonuniform thickness on the flow fields. Please refer to **Response G** for details. Motivated by that, Eq. (11) is proposed to provide an efficient way to analyze the effect of aquifer thickness on the flow field. To take advantage of the analytical solution in analyzing the thickness effect, a closed-form expression describing the varying thickness must be used. Three types of expressions have been used in the literature to describe the thickness of the aquifer, such as linear (Hantush, 1962a; Zamrsky et al., 2018), quadratic (Cuello et al., 2017; Zamrsky et al., 2018), and exponential (Hantush, 1962b; Marino and Luthin, 1982) varying thickness expressions.

b. Expanding Eq. (18) as a power series, it follows that

$$B(x) = \beta e^{\alpha x} = \beta \left[ 1 + \alpha x + \frac{1}{2!}(\alpha x)^2 + \frac{1}{3!}(\alpha x)^3 + \cdots \right]$$

It can be clearly seen that Eq. (18) contains not only a linear and a quadratic term, but also nonlinear (higher order) terms.

These are the reasons why a confined aquifer with varying thickness is approximated by an exponentially varying aquifer. However, this work does

provide a general framework (namely, Eqs. (11) or (17)) for evaluating flow fields in nonuniform aquifers with arbitrarily varying thickness.

Cuello, J. E., Guarracino, L., and Monachesi, L. B.: Groundwater response to tidal fluctuations in wedge-shaped confined aquifers. Hydrogeol. J., 25(5), 1509-1515, 2017.

Hantush, M. S.: Flow of ground water in sands of non-uniform thickness, part 2. Approximate theory, J Geophys. Res. 67(4), 711-720, 1962a.

Hantush, M. S.: Flow of ground water in sands of non-uniform thickness, part 3. Flow to wells, J. Geophys. Res., 67(4), 1527-1534, 1962b.

Marino, M.A., Luthin, J.N., 1982. Seepage and Groundwater, Elsevier, New York, 1982.

Zamrsky, D., Oude Essink, G. H., and Bierkens, M. F.: Estimating the thickness of unconsolidated coastal aquifers along the global coastline, Earth Syst. Sci. Data, 10(3), 1591-1603, 2018.

**F.** Applicability of this model: Are there any data to compare the results of this model to? This might be out of the scope of this work but it would be nice to see additional discussion related to the applicability of this model and its range of validity.

**Response**

 a. To our knowledge, modeling of the natural recharge-discharge process in confined aquifers of nonuniform thickness as a nonstationary process has not been previously presented in the literature. No data are available to compare the results of this work. An application of the proposed model to predict the outflow discharge is added on page 22 (Line 361) as

 "**4.2 Application in the prediction of outflow discharge**

 The usefulness of the stochastic theory presented here lies in its essentially predictive nature. The variance can be used as a quantification of the uncertainty associated with the prediction in field situations using the linear system model. In this sense, the solution of Eq. (1) $\pm$ two times the square root of the variance provides a rational framework for predicting discharge over a relatively large spatial scale where direct observations of such a dependent variable are not possible.

 For large times, the first term in Eq. (37) dominates the sum of the other terms, and therefore the transfer function can be approximated by

$$\left| \Lambda_q(t;\omega) \right|^2 = \frac{256}{\pi^2} L^2 \mathcal{G}^2 \left[ \pi \cos(\pi \Upsilon) - \mu \sin(\pi \Upsilon) \right]^2 \frac{1}{\Theta_1^2 + \Gamma^2} \frac{1}{\nabla^2 + \Gamma^2}$$

$$\left[ 1 + 2\nabla \Xi\, T_s + T_s^2 - 2(\nabla \Xi + T_s)\cos(\Omega_t) - 2\Xi\,\Omega_t \sin(\Omega_t) + \Xi^2 \right], \qquad (42)$$

where $\Theta_l = 1 + \pi^2 \eta^2$, $\nabla = \overline{K}\,\tau_0(\pi^2 + \mu^2)/(S_s L^2)$, $\Xi = \left[ \exp(-\Theta_l t/\tau_0) - T_s \right]/(\nabla - \Theta_l)$, and $T_s = \exp(-\nabla t/\tau_0)$. If the variation of the rainfall event is generated by a random white noise forcing, the variance of the outflow discharge at large times can then be calculated using Eq. (42) as

$$\sigma_q^2(t) = \int_{-\infty}^{\infty} S_{qq}(t;\omega)\,d\omega = \int_{-\infty}^{\infty} \left| \Lambda_q(t;\omega) \right|^2 S_{\xi\xi}(\omega)\,d\omega$$

$$= \frac{256}{\pi^2} L^2 \mathcal{G}^2 G_0 \left[ \pi \cos(\pi \Upsilon) - \mu \sin(\pi \Upsilon) \right]^2 \left\{ \frac{\Xi^2}{\Theta_1} + \frac{1 + 2\nabla \Xi\, T_s + T_s^2}{\nabla \Theta_1(\nabla + \Theta_1)} \right.$$

$$\left. -2(\nabla \Xi + T_s)\frac{\Theta_1 \cosh(\nabla t/\tau_0) - \nabla \cosh(\Theta_l t/\tau_0)}{\nabla \Theta_1^3 + \Theta_1 \nabla^3} - 2\Xi \frac{\sinh(\Theta_l t/\tau_0) - \sinh(\nabla t/\tau_0)}{\Theta_1^2 - \nabla^2} \right\}, \qquad (43)$$

where $G_0$ represents a constant spectral density of a white noise process. Note that white noise is a signal that contains all frequencies in equal proportions, that is, a signal whose spectrum is flat.

After observing the recharge rate $R(t)$ over time at the outcrop of the aquifer and identifying input parameters such as the specific storage coefficient, mean hydraulic conductivity and geometrical parameters of the aquifer and the characteristic time and length scales of the rainfall event for a given area or region, the discharge can be determined under uncertainty in the far downstream aquifer area, Eq. (1) together with Eq. (26) ± two times the square root of Eq. (43). It provides an important basis for the rational management of regional groundwater resources in complex geologic settings under uncertainty."

b. A note on the validity of the proposed model is added on page 17 (Line 290) as "Note that the linearity in modeling the recharge-discharge response of a catchment in Eq. (1), which was originally developed for large catchments, increases with catchment area (e.g., Chow et al., 1988). This implies that the impulse responses and transfer functions derived here are valid in large confined aquifers."

Chow, V. T., Maidment, D. R., and Mays, L. W.: Applied hydrology, McGraw-Hill, 1988.

**G.** Please highlight the key novelties of this work with respect to the previously published literature on recharge and aquifers.

**Response**

The main novelties of this work are: (1) the development of a diffusion equation that explicitly accounts for the thickness of the aquifer, (2) the modeling of the recharge-discharge process as a nonstationary process, and (3) the presentation of the expression for the transfer function of the linear confined aquifer system using a stochastic framework. They are highlighted on page 20 (Line 328) as

"The traditional approach to regional groundwater flow problems introduces the transmissivity term, the depth-integrated hydraulic conductivity operator

$$T(x_1, x_2) = \int_{b_1(x_1, x_2)}^{b_2(x_1, x_2)} K(x_1, x_2, x_3) \, dx_3 \tag{39}$$

into to the groundwater flow equation (diffusion equation) to reduce the three-dimensional equation to a two-dimensional one:

$$S(x_1, x_2) \frac{\partial}{\partial t} h(x_1, x_2, t) = \frac{\partial}{\partial x_i} [T(x_1, x_2) \frac{\partial}{\partial x_i} h(x_1, x_2, t)] \qquad i = 1, 2 \tag{40}$$

This means that the effects of both the variation of $K$ in $x_3$-direction and the aquifer thickness are implicitly reflected in the term $T(x_1, x_2)$. This leads to great difficulties in assessing the influence of aquifer thickness on the flow field with Eq. (40).

The proposed diffusion equation of this work,

$$S_s(x_1,x_2)\frac{\partial}{\partial x_i}\tilde{h}(x_1,x_2,t) = \frac{1}{B(x_1,x_2)}\frac{\partial}{\partial x_i}\left[K(x_1,x_2)B(x_1,x_2)\frac{\partial}{\partial x_i}\tilde{h}(x_1,x_2,t)\right] + K(x_1,x_2)\frac{\partial}{\partial x_i}\ln B(x_1,x_2)\frac{\partial}{\partial x_i}\tilde{h}(x_1,x_2,t) \quad i=1,2 \tag{41}$$

derived by the hydraulic approach (Bear, 1979; Bear and Cheng, 2010), provides an efficient way to analyze flow fields in confined aquifers of non-uniform thickness. Note that Eq. (41) is the reformulation of Eq. (11). In addition, the usual observations of flow in porous media are measurements of hydraulic head from wells screened over extended sections of the medium. The measurement at a given location approximately represents a depth-averaged actual hydraulic head resulting from flow through a three-dimensional hydraulic conductivity field across the thickness of the medium. This means that the depth-averaged head representation used in Eq. (41) is consistent with what is observed in the fields.

Climate changes have a direct influence on the rainfall event (e.g., Trenberth, 2011; Pendergrass et al., 2014; Eekhout et al., 2018). The

nonstationarity in the statistical properties of rainfall field is a representation of climate change (e.g., Razavi et al., 2015; López and Francés, 2013; Benoit et al., 2020). The nonstationary effect of climatic change over time on variability of groundwater specific discharge has not yet been well characterized in the. The transfer function in Eq. (37), which relates the nonstationary spectra of the rainfall fluctuations to those of integrated discharge variation, generalizes existing studies that considered stationary recharge/discharge fields. To our knowledge, it has not been previously presented in the literature and has the potential to analyze the effects of climate change on temporal groundwater specific discharge variability. "